# Tailored Physical Activity Interventions for Long COVID: Current Approaches and Benefits—A Narrative Review

**DOI:** 10.3390/healthcare12151539

**Published:** 2024-08-03

**Authors:** Guhua Jia, Chun-Hsien Su

**Affiliations:** 1Sports Teaching Department, Shanghai University of Medicine & Health Sciences, Shanghai 201318, China; 1800030@sumhs.edu.cn; 2Department of Exercise and Health Promotion, Chinese Culture University, Taipei City 111369, Taiwan; 3College of Kinesiology and Health, Chinese Culture University, Taipei City 111369, Taiwan

**Keywords:** exercise therapy, stamina, life quality, dyspnea, fatigue

## Abstract

This narrative review explores the essential role of physical activity in managing long COVID, which is characterized by persistent symptoms such as fatigue, breathlessness, and cognitive impairments following SARS-CoV-2 infection. In this context, “physical activity” includes various exercises, such as aerobic and resistance training, as well as flexibility and balance exercises, thereby encompassing the subset known as “exercise”. The methodology involved a comprehensive literature search across PubMed, EBSCO (EDS), and Google Scholar, selecting peer-reviewed articles from December 2019 to June 2024 focusing on long COVID physical activity interventions. The review highlights that tailored exercise programs, adjusted to individual health status and abilities, significantly alleviate symptoms and enhance psychological well-being. Key findings emphasize the importance of personalized exercise prescriptions due to the variability in patient responses and the need for a multidisciplinary approach in developing and monitoring interventions. Despite promising outcomes, the review identifies research gaps, including the need for long-term studies, randomized controlled trials, and deeper mechanistic insights. It suggests that standardized reporting, templates like the Consensus on Exercise Reporting Template (CERT), and integrating digital health tools are essential for optimizing interventions. Overall, the review advocates including personalized physical activity or exercise programs in standard care for long COVID to improve patient outcomes and quality of life.

## 1. Introduction

### 1.1. Background

Long COVID, or Post-Acute Sequelae of SARS-CoV-2 infection (PASC), is a multifaceted condition with a substantial global impact, characterized by persistent symptoms such as fatigue, breathlessness, and “brain fog” following the resolution of the acute phase of COVID-19, affecting a significant global population [1,2]. This syndrome leads to notable declines in physical and mental well-being, challenging individuals’ daily functioning and quality of life. The variability in symptoms necessitates personalized and multidisciplinary care approaches due to the impacts on physical function, mental health, and economic productivity [3,4,5]. The current understanding of the long-term pathophysiology of COVID-19 is limited, indicating a need for effective treatments and rehabilitation strategies. Research has highlighted physical activity as a critical area with potential benefits [6,7]. We acknowledge that “physical activity”, “exercise”, and “physical fitness” represent distinct concepts, although they are sometimes used interchangeably. Physical activity can be categorized into various daily activities, including occupational, sports, conditioning, household, or other activities. Exercise, a subset of physical activity, is defined as planned, structured, and repetitive activity to improve or maintain physical fitness. This study reviews the benefits of physical activity and exercise for individuals with long COVID. In this context, “physical activity” encompasses a range of exercises, including aerobic and resistance training, as well as flexibility and balance exercises, thereby incorporating its subset, “exercise” [7,8]. Tailoring these interventions to individual needs, considering the heterogeneity of long COVID symptoms, is essential for maximizing their efficacy [8,9].

Emerging evidence suggests that structured physical activity or exercise can mitigate some chronic symptoms of long COVID. For instance, regular aerobic exercise has been shown to reduce fatigue and improve cardiovascular endurance, while resistance training can enhance muscle strength and physical function [10,11]. Moreover, incorporating elements of flexibility and balance training can aid in reducing the risk of falls and improving overall mobility, which is particularly beneficial for those experiencing dizziness or muscle weakness [12]. Implementing physical activity interventions for long COVID patients poses several clinical challenges despite the potential benefits. These include determining the appropriate intensity and duration of exercise, managing symptom fluctuations, and addressing the psychological barriers to physical activity, such as fear of symptom exacerbation and lack of motivation [13,14]. Multidisciplinary approaches involving physiotherapists, occupational therapists, psychologists, and other healthcare professionals are crucial for developing and implementing effective rehabilitation programs [15]. The economic implications of long COVID are also significant, as prolonged illness can lead to reduced work capacity and increased healthcare costs. Effective rehabilitation strategies, including tailored physical activity interventions, can alleviate these economic burdens by enhancing patients’ functional abilities and facilitating their return to work [16,17,18,19].

Addressing the impacts of COVID-19 through tailored physical activity interventions is crucial for improving the health and quality of life of affected individuals amid the pandemic. This narrative review explores current approaches and clinical challenges in managing long COVID through tailored physical activity interventions to provide insights into effective rehabilitation strategies for this complex condition. It emphasizes the importance of individualized care and the integration of multidisciplinary support to address the diverse needs of long COVID patients.

### 1.2. Purpose and Scope

This narrative review aims to explore the role of physical activity in managing long-term COVID-19 symptoms and improving patient outcomes. Long COVID, characterized by persistent symptoms such as fatigue, breathlessness, and brain fog, poses significant challenges to healthcare providers and patients, impacting physical, mental, and social well-being. This review examines various exercise intensities, frequencies, and types through observational studies, clinical trials, systematic reviews, and qualitative research to assess their effectiveness in alleviating symptoms and enhancing well-being. Key aspects include the optimal duration and intensity of exercises, the benefits of different exercise forms, and the importance of personalized exercise programs tailored to individual needs. Additionally, it addresses the safety and tolerability of physical activities, emphasizing the need for tailored exercise prescriptions to minimize risks and maximize recovery benefits.

## 2. Relevant Collection of Research for Review

### 2.1. Search Strategy Description

A comprehensive literature search was carried out across several electronic databases, namely PubMed, EBSCO (EDS), and Google Scholar, to gather a broad spectrum of research on the intersection of physical activity and long COVID. The search strategy employed a combination of keywords such as “long COVID” and “Post-Acute Sequelae of SARS-CoV-2 infection (PASC)”, alongside terms linked to “physical activity”, “exercise”, “rehabilitation”, as well as specific modalities like “aerobic” and “strength training”, using Boolean operators for precision. This search covered research publications from the beginning of the COVID-19 pandemic in December 2019 to June 2024, ensuring the inclusion of the most recent findings. 

### 2.2. Roles of Authors and Conflict Resolution

Two reviewers searched for relevant articles to develop the objective of this study. Each author had specific roles, including initiating the search, screening the articles, extracting data, and drafting the manuscript sections. Any disagreements about study inclusion were resolved through discussion until a consensus was reached. If consensus could not be achieved, a third-party expert in exercise physiology was consulted to make the final decision.

### 2.3. Selection and Inclusion Criteria

Selection criteria were strictly adhered to, encompassing peer-reviewed English language articles that delved into physical activity interventions for long COVID sufferers. These included a range of study designs, from observational studies and clinical trials to qualitative research. Exclusions were made for content not directly related to long COVID, such as editorials, commentaries, and studies focusing solely on the acute phase of COVID-19 without considering the implications for long-term recovery, ensuring a focused and relevant collection of research for review. The initial selection process began with 216 studies, refined through three filters based on specific inclusion criteria. The Before-Screening Level removed 112 duplicates, leaving 104 studies. The Abstract Level applied criteria such as research methods, relevance to physical activity, exercise or COVID, open-access or peer-reviewed status, publication date (2019–2024), and English language, resulting in 47 studies. The Full-Text Level focused on systematic reviews, meta-analyses, detailed statistical measurements, and relevant review studies, excluding editorials and non-research articles. This stage resulted in 18 studies for the comprehensive review.

The review encompassed diverse study designs, including observational studies, clinical trials, systematic reviews, meta-analyses, and qualitative research. Participant demographics varied widely, covering different ages, genders, and stages of long COVID. Physical activity interventions ranged from low-intensity exercises to vigorous activities. Outcome measures included symptom improvement, quality of life, physical function, and psychological well-being, providing a comprehensive overview of physical activity’s impact on long COVID.

### 2.4. Each Reference’s Evidence Level

The Agency for Healthcare Research and Quality (AHRQ) provides guidelines for evidence-based practice, including a hierarchy of evidence that similarly places systematic reviews and meta-analyses at the top, followed by randomized controlled trials and cohort studies, etc. Each reference’s evidence level is graded as A: systematic reviews and meta-analyses; B: randomized controlled trials (RCTs); C: cohort studies, case-control studies, cross-sectional surveys, case studies, and/or observational studies; D: review or evidence insufficient for categories A, B or C.

## 3. Effects of Physical Activity or Exercise on Long COVID Symptoms

Findings consistently demonstrated positive effects of physical activity on long COVID symptoms. Participants highlighted the importance of outdoor spaces for physical activity and the impact of changes in daily routines and COVID-19 restrictions on their ability to be active. Numerous participants mentioned risks or threats to participation, such as fear of contracting COVID-19. Both physical and mental health were highlighted as essential motivators for physical activity, with numerous participants using technology to support their active lifestyle [20]. Moderate long-term exercise training might be an effective strategy to reduce the likelihood of severe presentation of COVID-19. Improved resistance to infection can be seen in two ways, one of which involves influencing the immune-regulatory pathway (IL-6/cortisol) [21]. Articles revealed a significant association between sufficient physical activity and reduced COVID-19 hospitalization and mortality. The analysis showed a weighted odds ratio of 0.541 for hospitalization and 0.61 for mortality, indicating a protective effect of physical activity [22]. Regular physical activity may reduce COVID-19-related hospitalization and mortality through several physiological mechanisms. Exercise boosts immune function by enhancing the circulation of immune cells, improving the body’s infection response. It also has anti-inflammatory effects, mitigating severe inflammatory responses seen in COVID-19. Improved cardiovascular health from regular exercise helps prevent complications like myocarditis and blood clots. Physical activity regulates blood glucose levels and improves insulin sensitivity, reducing the risk of comorbidities like diabetes and obesity. Additionally, exercise strengthens respiratory muscles, aiding in maintaining adequate oxygen levels and reducing respiratory complications. By managing chronic conditions such as hypertension and obesity, regular physical activity decreases the prevalence of comorbidities linked to severe COVID-19. Exercise also reduces stress and anxiety, enhances immune function, and improves blood circulation, supporting efficient nutrient and oxygen delivery. These mechanisms collectively contribute to reduced risk of severe outcomes, highlighting the importance of an active lifestyle in combating COVID-19 [22,23].

Early physical rehabilitation interventions applied to COVID-19 patients who were discharged from the hospital improved multiple parameters related to functional capacity, pulmonary function, quality of life, and mental health status [23,24,25]. Regular moderate-intensity exercise was associated with significant energy and endurance improvements for fatigue, a common and debilitating symptom. The study found that physical activity during the pandemic was associated with a reduced likelihood of long COVID and a shorter duration of symptoms. Those who remained physically active from before to during the pandemic were less likely to report long COVID, fatigue, neurological complications, cough, and loss of sense of smell or taste [26,27].

Physical activity programs could help counteract muscle atrophy and fatigue observed in long COVID patients [28]. Breathlessness also saw improvement through tailored aerobic exercises that enhanced lung capacity and efficiency. A review of articles accessed through electronic databases found that exercise interventions and pulmonary rehabilitation have shown promise in improving functional exercise capacity, dyspnea, and fatigue in people with long COVID [27,29]. Additionally, strength and flexibility exercises contributed to alleviating joint and muscle pain. In contrast, targeted breathing exercises helped manage respiratory symptoms and anxiety, showcasing the multifaceted benefits of exercise on long COVID recovery [13,30,31]. Table 1 in our research concisely summarizes the effects of physical activity or exercise on long COVID symptoms.

## 4. Variability in Response to Physical Activity or Exercise Interventions

This review highlighted the variability in response to physical activity interventions, noting that individual outcomes could differ based on several factors, including baseline severity of symptoms, pre-existing conditions, and the specific type and intensity of physical activity undertaken. Physical activity plays a crucial role in preventing and treating COVID-19, promoting the recovery of bodily function, alleviating post-acute COVID-19 syndromes, and improving psychological well-being. It is recommended that appropriate exercise prescriptions for different populations be developed under the guidance of a physician [7].

Some studies reported that patients with more severe long COVID symptoms at baseline experienced more significant improvements with tailored exercise programs. For instance, a study comparing high-intensity interval training (HIIT) and moderate-intensity training (MIT) found that both were effective in improving mental well-being during COVID-19 confinement, with HIIT being more beneficial for reducing depression [32]. Conversely, other studies found that mild- to moderate-intensity exercises were more useful for individuals with less severe symptoms, emphasizing the need for personalized exercise prescriptions. HIIT triggers a significant release of endorphins and increases levels of serotonin and norepinephrine, improving mood and reducing depression. It also elevates brain-derived neurotrophic factor (BDNF) levels, supports neuronal growth, and enhances cognitive function. HIIT leads to a pronounced reduction in cortisol levels, improves sleep quality, reduces systemic inflammation, and enhances heart rate variability (HRV), contributing to better autonomic nervous system function. The challenging nature of HIIT provides a greater sense of accomplishment, further boosting mental health. These mechanisms explain HIIT’s superior effectiveness in reducing depression and enhancing mental well-being compared to MIT [32,33].

A systematic review of seven studies revealed that resistance and aerobic exercise programs can improve functional capacity and quality of life in post-COVID-19 patients. These studies demonstrated improvements in muscle strength, functional capacity, fatigue, and quality of life following exercise interventions [33,34]. For a detailed overview of the variability in response to physical activity or exercise interventions in long COVID, please refer to Table 2.

## 5. Safety and Feasibility of Physical Activity or Exercise Interventions

Overall, physical activity interventions were safe and feasible for individuals with long COVID, with few adverse events reported. Most studies emphasized the importance of gradually increasing the intensity and duration of exercise to avoid exacerbating symptoms. Physical activity may have a protective effect against fatal outcomes in patients with COVID-19. The protective effect of physical activity on COVID-19 outcomes may be attributed to specific types of exercise, such as resistance and endurance exercises. Further research is needed to understand the biological mechanisms behind these findings [35]. Physical exercise has been shown to modulate interferon responses and innate immune cell activity, which are crucial in the initial defense against viral infections. By enhancing the immune system and improving overall health, physical exercise may serve as a valuable complementary tool in preventing and treating COVID-19. In addition to its immunomodulatory effects, exercise can help control inflammation, oxidative stress, and nitric oxide synthesis, contributing to a comprehensive approach to combating the virus [36]. This review underscored the necessity of closely monitoring individuals during physical activity interventions, particularly those with severe long-term COVID or underlying health conditions, to ensure their safety and optimize the benefits of exercise. A meta-analysis of existing literature was conducted, analyzing various outcome measures related to physical function in COVID-19 patients. The results indicated that physical activity interventions effectively improved the 6 min Walk Test, 30 s Sit-to-Stand Test, Timed Up-and-Go test, Forced Vital Capacity, and Forced Expiratory Volume in COVID-19 patients. Overall, physical activity was found to play a crucial role in enhancing exercise capacity and pulmonary function in COVID-19 patients, promoting their overall physical health and recovery. COVID-19 patients must undergo an accurate physical assessment before physical activity to ensure their safety and well-being [7,37]. Table 3 highlights the safety and feasibility of physical activity or exercise interventions for individuals recovering from long COVID, underscoring the importance of gradual progression and close monitoring to minimize risks and enhance recovery outcomes.

## 6. Discussion

### 6.1. Effects of Physical Activity or Exercise on Long COVID Symptoms

The review provides a comprehensive synthesis of evidence indicating the beneficial effects of physical activity on managing long COVID symptoms. Tailored exercise programs or physical activity have demonstrated substantial improvements in patients with severe long COVID symptoms at baseline, emphasizing the importance of personalized exercise prescriptions [13]. Mild- to moderate-intensity exercises, on the other hand, have shown more significant benefits for individuals with less severe symptoms [26]. These findings highlight the necessity of individualized exercise regimens in rehabilitating long COVID patients, enabling targeted interventions that address specific needs, thereby enhancing functional capacity and quality of life [38]. Personalized exercise prescriptions consider factors such as obesity and vaccination status, which influence the response to rehabilitation [39]. Case studies, such as those by Torres et al., illustrate significant improvements in lung function, cardiorespiratory fitness, endurance capacity, and muscle strength following exercise interventions using FITT-VP (Frequency, Intensity, Time, Type, Volume, and Progression) principles and the Consensus on Exercise Reporting Template (CERT) [40]. The Consensus on Exercise Reporting Template (CERT) is a standardized tool designed to improve the reporting quality of exercise interventions in clinical research. The CERT, with its checklist of 16 items, promotes transparency by covering critical aspects of exercise interventions, including a detailed description of the exercise intervention, setting and supervision, delivery, adherence and compliance, tailoring and modifications, training of providers, and safety considerations. Using CERT, researchers can provide a comprehensive and transparent account of the exercise intervention, facilitating better understanding, reproducibility, and application in practice [41]. Similarly, Binetti et al. reported enhanced 6 min walk test results and reduced fatigue levels in long COVID patients participating in supervised exercise programs, with biomarkers like creatinine potentially predicting rehabilitation response [26]. Rehabilitation interventions have also improved lung function, exercise capacity, dyspnea severity, quality of life, and mental health outcomes [42].

### 6.2. Variability in Response to Physical Activity or Exercise Interventions

The review highlights considerable variability in individual responses to physical activity interventions, underscoring the need for personalized exercise prescriptions. Patients with more severe long COVID symptoms at baseline experienced more significant improvements with tailored exercise programs, including high-intensity interval training (HIIT) and moderate-intensity training (MIT). Conversely, mild to moderate-intensity exercises benefited individuals with less severe symptoms. HIIT and MIT can enhance mitochondrial function and efficiency for patients with severe symptoms, improving energy production and reducing fatigue. HIIT, in particular, stimulates greater production of endorphins and neurotransmitters such as serotonin and norepinephrine, which help in alleviating depression and anxiety, which is common in long COVID [7,32]. Moderate-intensity exercises benefit those with less severe symptoms by improving cardiovascular and respiratory function and enhancing oxygen delivery and utilization throughout the body [7,26,33,34]. These exercises also help regulate immune function and reduce systemic inflammation, which can mitigate the chronic inflammatory state associated with long-term COVID-19. The improved autonomic function, as evidenced by enhanced heart rate variability (HRV), further supports the reduction of symptoms like breathlessness and fatigue [7]. Factors such as baseline severity of symptoms, pre-existing conditions, and the specific physical activity type and intensity are crucial in determining individual outcomes. The role of healthcare professionals in developing these personalized exercise prescriptions is vital, providing the audience with a sense of support and care. The Borg Scale, also known as the Rating of Perceived Exertion (RPE) scale, is a numerical scale that measures an individual’s perceived effort during physical activity. It assesses how hard a person feels they are working by taking into account factors such as heart rate, respiration rate, sweating, and muscle fatigue [43]. This subjective measure is crucial for tailoring exercise intensity to individual capabilities and ensuring safety. The 1RM (one-repetition maximum) is the maximum weight a person can lift for one complete repetition of a given exercise. It is a standard measure used in strength training and exercise physiology to assess the maximum strength of an individual’s muscles [44]. Utilizing 1RM helps design strength training programs that align with an individual’s capacity, preventing overtraining and promoting gradual strength improvements. Personalized exercise prescriptions considering individual baseline conditions, symptom severity, and specific physiological responses are essential for effectively managing long COVID. Tailored interventions, supported by tools like the Borg Scale and 1RM, can optimize health outcomes and improve the quality of life for affected individuals.

### 6.3. Feasibility and Safety of Physical Activity or Exercise Interventions

Physical activity interventions were generally safe and feasible for individuals with long COVID, with few adverse events reported. Most studies emphasized the importance of gradually increasing the intensity and duration of exercise to avoid exacerbating symptoms [35]. Physical activity may protect against fatal outcomes in COVID-19 patients, possibly due to specific types of exercise, such as resistance and endurance training [35]. The necessity of closely monitoring individuals during physical activity interventions, particularly those with severe long COVID or underlying health conditions, is highlighted to ensure safety and optimize the benefits of exercise [7,37]. Monitoring helps prevent overexertion and ensures the exercise intensity is appropriate for the individual’s health. Several mechanisms support the feasibility and safety of these interventions. Exercise can modulate interferon responses, which is critical in the antiviral defense mechanism, enhancing the body’s immune function by increasing the activity of natural killer cells and T cells [45]. This heightened immune response helps manage viral infections more effectively [36]. Additionally, exercise controls inflammation by reducing levels of pro-inflammatory cytokines and increasing anti-inflammatory cytokines, which can help manage the chronic inflammation seen in long COVID [46]. Furthermore, exercise reduces oxidative stress by enhancing the body’s antioxidant defenses, which prevents cellular damage and promotes overall recovery [36]. Moreover, exercise enhances nitric oxide synthesis, improving vascular health by promoting vasodilation, which leads to better blood flow and oxygen delivery to tissues [47]. This vascular improvement is particularly beneficial for individuals with long COVID, who may experience compromised blood flow and oxygenation. Moreover, physical activity enhances exercise capacity and pulmonary function, promoting overall physical health and recovery in COVID-19 patients. Improved pulmonary function is essential for those with long COVID, as it can help alleviate symptoms such as breathlessness and improve oxygen saturation levels. Enhanced exercise capacity contributes to better physical endurance and strength, critical for daily functioning and overall quality of life. 

The physiological benefits of exercise, including improved immune function, reduced inflammation, enhanced antioxidant defenses, and better vascular health, support the feasibility and safety of physical activity interventions for individuals with long COVID. Gradual progression in exercise intensity and careful monitoring are essential to maximize these benefits and ensure the well-being of participants.

### 6.4. Clinical Implications

Incorporating physical activity as a critical component of therapeutic management for long-term COVID-19 is essential. Tailored exercise programs should be recommended based on symptom severity, baseline physical condition, and personal preferences, as patient engagement and adherence are crucial for the success of the program [48,49,50]. A multidisciplinary approach involving physiotherapists, exercise physiologists, and mental health professionals is vital for comprehensively addressing the complex needs of long-term COVID patients [24,40,51,52]. Despite the lack of standardization in reporting exercise interventions, templates like CERT and FITT-VP exercise prescription principles show promise in developing best practice guidelines [13,53,54]. Rehabilitation interventions such as endurance, flexibility, strength training, pulmonary rehabilitation, task-specific exercises, psychological rehabilitation, and pain management can significantly improve functional capacity and quality of life in long COVID patients [26,55,56,57]. Cardiopulmonary exercise testing (CPET) and pulmonary rehabilitation are particularly effective in managing dyspnea and muscle fatigue [41,58]. CPET helps diagnose dyspnea, exercise intolerance, and cardiovascular and pulmonary diseases and aids in developing personalized exercise programs. It also provides prognostic information and supports preoperative assessments and research. Generally safe under medical supervision, CPET is conducted in controlled environments for immediate medical assistance if needed.

### 6.5. Research Gaps and Future Directions

Despite the promising evidence, several research gaps remain. More significant, well-designed, randomized controlled trials are needed to establish causal relationships between specific types and intensities of physical activity and improvements in long COVID symptoms. Longitudinal studies are essential to understand the long-term effects of exercise interventions and identify potential delayed adverse outcomes, such as overuse injuries or exacerbation of pre-existing conditions. Further research should explore the underlying mechanisms of physical activity benefits to develop targeted interventions and provide insights into optimizing rehabilitation strategies. This review has limitations, including potential biases in study selection and reliance on published literature, which may influence the findings. The heterogeneity of the studies in design, population, and interventions complicates the direct comparison of results. Additionally, the rapidly evolving nature of long COVID research necessitates ongoing updates to the evidence base, as newer studies may provide further insights and refine current knowledge.

## 7. Conclusions

This review emphasizes the pivotal role of physical activity in managing long COVID, a condition marked by persistent symptoms such as fatigue, breathlessness, and muscle pain following an initial SARS-CoV-2 infection. Tailored exercise programs, customized to each individual’s health status and abilities, have shown significant potential in alleviating these symptoms and enhancing psychological well-being. The variability in patient responses to physical activity highlights the need for personalized exercise prescriptions considering symptom severity, pre-existing health conditions, and prior physical activity levels. The findings underscore the importance of a flexible, patient-centered approach, supported by a multidisciplinary team of healthcare professionals, to develop and monitor effective and safe exercise interventions. 

Future research should prioritize long-term studies, well-designed randomized controlled trials, and mechanistic investigations to deepen the understanding of how physical activity benefits long COVID patients. Additionally, standardized reporting using templates like CERT, identifying predictors of rehabilitation response, and integrating digital health tools are crucial for refining these interventions. Overall, this review advocates integrating personalized physical activity programs into the standard care for long COVID. It emphasizes their potential to significantly improve patient outcomes and quality of life while calling for continued research to address existing gaps and optimize rehabilitation strategies.

## Figures and Tables

**Table 1 healthcare-12-01539-t001:** Summary of the effects of physical activity or exercise on long COVID symptoms.

Symptom	Type of Exercise	Effect	Reference: Grade ^1^
Fatigue	Moderate-intensity exercise	Significant improvements in energy and endurance. Reduction in fatigue scores, increased exercise capacity, enhanced daily activity levels, improved muscle strength, better quality of sleep, decreased perceived exertion, and enhanced mental well-being.	[26]: D [27]: C [28]: D
Breathlessness	Tailored aerobic exercises	Enhanced lung capacity and efficiency. Increased forced vital capacity, increased forced expiratory volume, improved peak expiratory flow rate, reduced breathlessness scores, improved exercise tolerance, and increased maximal voluntary ventilation.	[27]: C [29]: D
Joint and Muscle Pain	Strength and flexibility exercises	Alleviation of joint and muscle pain. Reduced pain scores, increased range of motion, improved muscle strength, enhanced functional mobility, decreased stiffness scores, and increased daily activity levels.	[13]: D [30]: C [31]: A
Respiratory Symptoms	Targeted breathing exercises	Management of respiratory symptoms and anxiety. Reduced breathlessness scores, improved lung function, decreased respiratory rate, and improved anxiety scores.	[13]: D [30]: C [31]: A
Mental Health	Moderate-intensity exercise, Technology-supported physical activity	Improved mental health and motivation. Reduced anxiety scores, reduced depression scores, improved mood scores, enhanced self-reported motivation levels, and increased overall well-being scores.	[20]: C
Functional Capacity	Early physical rehabilitation interventions	Increased exercise tolerance, enhanced muscle strength, improved balance and stability, increased range of motion, improved functional mobility, and enhanced daily activity levels.	[23]: A [24]: C [25]: A
Pulmonary Function	Early physical rehabilitation interventions	Improvement in pulmonary function. Increased forced vital capacity, increased forced expiratory volume, improved peak expiratory flow rate, enhanced oxygen saturation levels, and improved maximal voluntary ventilation.	[23]: A [24]: C [25]: A
Quality of Life	Early physical rehabilitation interventions	Enhanced quality of life. Improved scores on quality of life questionnaires, increased vitality and energy levels, reduced pain scores, and improved performance in activities of daily living (adls).	[23]: A [24]: C [25]: A
Neurological Complications	General physical activity	Reduced likelihood of neurological complications. Improved cognitive function scores, reduced incidence of neurological symptoms (e.g., headaches, dizziness), improved balance and coordination, and lowered inflammation markers (e.g., c-reactive protein, interleukin-6 levels).	[26]: D [27]: C

^1^ Each reference’s evidence level is graded as A: systematic reviews and meta-analyses; C: cohort studies, case-control studies, cross-sectional surveys, case studies, and/or observational studies; D: review or evidence insufficient for categories A or C.

**Table 2 healthcare-12-01539-t002:** Variability in response to physical activity or exercise interventions in long COVID.

Factor	Details	Reference: Grade ^1^
Baseline Severity of Symptoms	Patients with more severe long COVID symptoms at baseline experienced more significant improvements with tailored exercise programs.	[7]: D
Type of Exercise	High-Intensity Interval Training (HIIT) and Moderate-Intensity Training (MIT) improved mental well-being, with HIIT being more beneficial for reducing depression during COVID-19 confinement.	[32]: B
Intensity of Exercise	Mild to moderate-intensity exercises were more beneficial for individuals with less severe symptoms, emphasizing the need for personalized exercise prescriptions.	[7]: D
Exercise Programs	Resistance and aerobic exercise programs improved functional capacity and quality of life in post-COVID-19 patients, demonstrating improvements in muscle strength, functional capacity, fatigue, and quality of life.	[33]: A [34]: A
Personalized Exercise Prescriptions	Personalized exercise prescriptions are essential, developed under the guidance of a physician to cater to different populations and individual needs.	[7]: D
Mental Well-being	Exercise interventions, including HIIT and MIT, improved mental well-being during confinement, demonstrating the psychological benefits of physical activity.	[32]: B
Functional Capacity and Quality of Life	Exercise programs showed significant improvements in muscle strength, functional capacity, fatigue reduction, and overall quality of life in post-COVID-19 patients.	[33]: A [34]: A

^1^ Each reference’s evidence level is graded as A: systematic reviews and meta-analyses; B: randomized controlled trials (RCTs); D: review or evidence insufficient for categories A or B.

**Table 3 healthcare-12-01539-t003:** Safety and feasibility of physical activity or exercise interventions in long COVID.

Aspect	Details	Reference: Grade ^1^
Safety and Feasibility	Physical activity interventions were found to be safe and feasible for individuals with long COVID, with few adverse events reported.	[35]: A
Gradual Progression	It is important to gradually increase the intensity and duration of exercise to avoid exacerbating symptoms.	[35]: A
Protective Effects	Physical activity may protect against fatal outcomes in patients with COVID-19, attributed to specific types of exercise such as resistance and endurance.	[35]: A
Immunomodulatory Effects	Physical exercise modulates interferon responses and innate immune cell activity, enhancing the initial defense against viral infections.	[36]: D
Control of Inflammation	Exercise helps control inflammation, oxidative stress, and nitric oxide synthesis, contributing to a comprehensive approach to combating the virus.	[36]: D
Close Monitoring	There is a need to closely monitor individuals during physical activity interventions, particularly those with severe long-term COVID or underlying health conditions.	[7]: D [37]: A
Improvement in Physical Function	Physical activity interventions improved the 6 min Walk Test, 30 s Sit-to-Stand Test, Timed Up-and-Go test, Forced Vital Capacity, and Forced Expiratory Volume.	[7]: D [37]: A
Enhancement of Exercise Capacity	Physical activity enhances exercise capacity and pulmonary function in COVID-19 patients, promoting overall physical health and recovery.	[7]: D [37]: A
Pre-activity Physical Assessment	COVID-19 patients must undergo an accurate physical assessment before physical activity to ensure safety and well-being.	[7]: D [37]: A

^1^ Each reference’s evidence level is graded as A: systematic reviews and meta-analyses; or D: review or evidence insufficient for categories A.

## Data Availability

Not applicable.

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
