# Peer review of "Tailored Physical Activity Interventions for Long COVID: Current Approaches and Benefits—A Narrative Review"

_healthcare, 2024, doi:10.3390/healthcare12151539_

Round 1

Reviewer 1 Report

Comments and Suggestions for Authors

Long covid sequale has been found to have a negative influence in peoples life and daily living. The need of a personalised approach is needed and this is well established. The authors have tried to present all data regarding  benefits and challenges of this. 

The authors named their review as a narrative one, yet their thorough work could allow them to present it as a scoping one.

1. There is need for a clear definition of the terms physical activity and exercise as often there is a confusement between them (Caspersen CJ, Powell KE, Christenson GM. Physical activity, exercise, and physical fitness: definitions and distinctions for health-related research. Public Health Rep. 1985;100(2):126-131.). Is their work clearly on physical activity or tailored exercise? Even from table 1, only ref 27 provided data on physical activity. Table 2 has a headline regarding physical activity, yet it presents exercise characteristics. It is confusing when authors name physical activity exercise programs that have spesific structure, and thus it is often confusing to tell one from another.

2. The authors included in their aims the need to present the importance of tailored approches. It is of hight importnace nont only to present data that supports this, but to give clinical information on how this could be done. There is a scarce note to CERT (please explain in abstract), but with out giving a more detailed description which is most significant for clinicians. And what about the use of borg scale or 1RM or event CPET. These should be presented in the findings and also descussed thoroughly.

3. Discussion is superficial. The authors did find differences regarding the most appropriate tailored program and this is important. But they need to present the physiological mechanisms that support this. Explaining means giving scientific evidence that will support clinicians to adopt their work.

The authors have made a significant work, with a high clinical importance. They need to better support their findings.

Author Response

healthcare-3109667 Reviewer 1

Long covid sequale has been found to have a negative influence in peoples life and daily living. The need of a personalised approach is needed and this is well established. The authors have tried to present all data regarding  benefits and challenges of this. 

The authors named their review as a narrative one, yet their thorough work could allow them to present it as a scoping one.

  1. There is need for a clear definition of the terms physical activity and exercise as often there is a confusement between them (Caspersen CJ, Powell KE, Christenson GM. Physical activity, exercise, and physical fitness: definitions and distinctions for health-related research. Public Health Rep. 1985;100(2):126-131.). Is their work clearly on physical activity or tailored exercise? Even from table 1, only ref 27 provided data on physical activity. Table 2 has a headline regarding physical activity, yet it presents exercise characteristics. It is confusing when authors name physical activity exercise programs that have spesific structure, and thus it is often confusing to tell one from another.

Thank you for your feedback. We acknowledge that "physical activity," "exercise," and "physical fitness" represent distinct concepts, although they are sometimes used interchangeably. Physical activity can be categorized into various daily activities, including occupational, sports, conditioning, household, or other activities. Exercise, a subset of physical activity, is defined as planned, structured, and repetitive activity to improve or maintain physical fitness.

In this context, "physical activity" encompasses a range of exercises, including aerobic and resistance training, as well as flexibility and balance exercises, thereby incorporating its subset, "exercise."

This study reviews the benefits of physical activity and exercise on long COVID. We agree that incorporating "exercise" into the subtitle and framework would provide more clarity and precision.

Please see lines 11-13 on page 1, 40-48 on pages 1-2, 185 on page 5, and 220 on page 6.

  1. The authors included in their aims the need to present the importance of tailored approches. It is of hight importnace nont only to present data that supports this, but to give clinical information on how this could be done. There is a scarce note to CERT (please explain in abstract), but with out giving a more detailed description which is most significant for clinicians. And what about the use of borg scale or 1RM or event CPET. These should be presented in the findings and also descussed thoroughly.

Thank you for your insightful feedback. We have addressed your concerns by providing additional information and clarifying certain aspects of our study.

The Consensus on Exercise Reporting Template (CERT) is a standardized tool designed to improve the reporting quality of exercise interventions in clinical research. With its checklist of 16 items, CERT promotes transparency by covering critical aspects of exercise interventions, including a detailed description of the exercise intervention, setting and supervision, delivery, adherence and compliance, tailoring and modifications, training of providers, and safety considerations. Using CERT, researchers can provide a comprehensive and transparent account of the exercise intervention, facilitating better understanding, reproducibility, and application in practice.

The Borg Scale, also known as the Rating of Perceived Exertion (RPE) scale, is a numerical scale that measures an individual's perceived effort during physical activity. Developed by Swedish psychologist Gunnar Borg, it helps assess how hard a person feels they are working, considering factors such as heart rate, respiration rate, sweating, and muscle fatigue.

Regarding using the Borg Scale, 1RM (one-repetition maximum), and CPET (cardiopulmonary exercise testing), we have incorporated these tools into our findings and discussed their significance thoroughly. These tools are essential for assessing exercise intensity and capacity, ensuring that tailored exercise interventions are appropriately calibrated to individual needs, enhancing their effectiveness and safety for patients with long COVID.

Cardiopulmonary exercise testing (CPET) evaluates the cardiovascular, pulmonary, and muscular systems during exercise to diagnose and manage medical conditions. Conducted on a treadmill or cycle ergometer with increasing intensity, CPET monitors ventilation, heart rate, blood pressure, ECG, oxygen saturation, and perceived exertion. Key measures include VO2 max, anaerobic threshold (AT), and ventilatory efficiency (VE/VCO2 slope), indicating aerobic fitness, metabolic shifts, and pulmonary function. CPET helps diagnose dyspnea, exercise intolerance, and cardiovascular and pulmonary diseases and aids in developing personalized exercise programs. It also provides prognostic information and supports preoperative assessments and research. Generally safe under medical supervision, CPET is conducted in controlled environments for immediate medical assistance if needed.

In the abstract, we have now included a brief explanation of CERT to ensure its relevance and application are apparent to clinicians.

Please see lines 22 on page 1, 270-277 on page 8, 303-317 on pages 8-9, 365-369 on page 10.

  1. Discussion is superficial. The authors did find differences regarding the most appropriate tailored program and this is important. But they need to present the physiological mechanisms that support this. Explaining means giving scientific evidence that will support clinicians to adopt their work.

The authors have made a significant work, with a high clinical importance. They need to better support their findings.

Thank you for your valuable feedback. We appreciate your recognition of the clinical importance of our work. In response to your suggestions, we have revised the discussion section to explain the physiological mechanisms that support the effectiveness of tailored exercise programs for long COVID.

Please see lines 150-164 on page 4, 205-213 on page 6, 289-300 on page 8, and 327-338 on page 9.

Reviewer 2 Report

Comments and Suggestions for Authors

Abstract:

Don't apply abbreviation for first one in the text. What means CERT?

Set keywords according to MeSH in PubMed central.

Introduction

"Physical activity, encompassing a range of exercises from aerobic and resistance training to flexibility and balance exercises, has improved cardiovascular health, 39 muscle strength, mental health, and overall quality of life". Include references for these sentences.

"Addressing COVID-19 impacts through tailored physical activity interventions is 61 crucial for improving affected individuals' health and quality of life amid the pandemic". Correct these sentences grammatically

Method

"December 2019 to the latest available research".  When was the Latest date of your comprehensive search?

Results:

Add subheading of results to your paper.

Write how many studies were identified in the searches. How many were excluded and finally selected?

How many reviewers was carried search for relevant articles to develop the objective of this study?

How grading papers in tables (Grad C, D,….).

"Many mentioned risks or threats to participation, such as fear of con- 118 tracting COVID-19" Many is unclear in this sentence.

Discussion:

" The review provides a comprehensive synthesis of evidence indicating that physical activity can significantly manage long COVID symptoms" change it to the effect of physical activity on …..and remove significantly.

Discussion divided to 3 parts based on results: first describe Effects of Physical Activity on symptoms fatigue, breathlessness, Variability in Response to Physical Activity Interventions and feasibility and safety….it is confusing for readers.

Comments on the Quality of English Language

 Minor editing of English language required

Author Response

healthcare-3109667 Reviewer 2

Don't apply abbreviation for first one in the text. What means CERT?

Set keywords according to MeSH in PubMed central.

Thank you for your feedback. We have made the following changes based on your suggestions:

  1. We have ensured that the first mention of the term is not abbreviated: Consensus on Exercise Reporting Template (CERT).

Please see line 22 on page 1.

  1. The keywords have been updated according to MeSH terms in PubMed Central:

Deleted: post-acute sequelae of SARS-CoV-2, rehabilitation, symptom management

Added: stamina (for physical endurance), life quality (for quality of life), dyspnea (for breathlessness), and fatigue

Thank you for your valuable input.

Please see line 26 on page 1.

Introduction

"Physical activity, encompassing a range of exercises from aerobic and resistance training to flexibility and balance exercises, has improved cardiovascular health, 39 muscle strength, mental health, and overall quality of life". Include references for these sentences.

Thank you for your feedback. We have revised the introduction to include references for the benefits of physical activity.

Please see line 48 on page 2.

"Addressing COVID-19 impacts through tailored physical activity interventions is 61 crucial for improving affected individuals' health and quality of life amid the pandemic". Correct these sentences grammatically

 Thank you for your feedback. We have revised the sentence for grammatical accuracy. The updated text is as follows:

"Addressing the impacts of COVID-19 through tailored physical activity interventions is crucial for improving the health and quality of life of affected individuals amid the pandemic."

Please see lines 69-71 on page 2.

Method

"December 2019 to the latest available research".  When was the Latest date of your comprehensive search?

 Thank you for your feedback. The comprehensive search covered the period from December 2019 to the latest available research as of June 2024.

Please see lines 15 on page 1 and 98 on page 3.

Results:

Add subheading of results to your paper.

Write how many studies were identified in the searches. How many were excluded and finally selected?

How many reviewers was carried search for relevant articles to develop the objective of this study?

Thank you for your feedback. We have incorporated the suggested results into the paper under the section "2. Relevant Collection of Research for Review," with subheadings "2.1. Search Strategy Description," "2.2. Roles of Authors and Conflict Resolution," "2.3. Selection and Inclusion Criteria," and "2.4. Each Reference’s Evidence Level."

The initial selection process began with 216 studies. After applying our selection criteria, 198 studies were excluded, resulting in 18 studies for the comprehensive review. The search for relevant articles to develop the objective of this study was carried out by two reviewers.

Please see lines 98-121 on page 9.

How grading papers in tables (Grad C, D,….).

The Agency for Healthcare Research and Quality (AHRQ) provides guidelines for evidence-based practice, including a hierarchy of evidence that similarly places systematic reviews and meta-analyses at the top, followed by randomized controlled trials and cohort studies. Each reference’s evidence level is graded as A: systematic reviews and meta-analyses; B: randomized controlled trials (RCTs); C: cohort studies, case-control studies, cross-sectional surveys, case studies, and/or observational studies; D: review or evidence insufficient for categories A, B or C.

Please see lines 130-136 on page 9.

"Many mentioned risks or threats to participation, such as fear of con- 118 tracting COVID-19" Many is unclear in this sentence.

 Thank you for your feedback. We have rewritten the sentence for clarity. The updated text is as follows:

"Numerous participants mentioned risks or threats to participation, such as fear of contracting COVID-19."

Please see line 141-142 on page 3.

Thank you for your valuable input.

Discussion:

" The review provides a comprehensive synthesis of evidence indicating that physical activity can significantly manage long COVID symptoms" change it to the effect of physical activity on …..and remove significantly.

Thank you for your feedback. We have revised the sentence as suggested. The updated text is as follows:

"The review provides a comprehensive synthesis of evidence indicating the effect of physical activity on managing long COVID symptoms."

Please see lines 256-257 on page 7.

Discussion divided to 3 parts based on results: first describe Effects of Physical Activity on symptoms fatigue, breathlessness, Variability in Response to Physical Activity Interventions and feasibility and safety….it is confusing for readers.

Thank you for your feedback. We have reorganized the discussion section to make it easier to read.

Please see lines 255 on page 7, 270-277 on page 8, 283-317 on pages 8-9, 318-350 on page 9, 351 on page 9, and 366 on page 10.

Round 2

Reviewer 1 Report

Comments and Suggestions for Authors

The authors have made a significant work and have well and thoroughly addressed all comments. They have enriched their manuscript and presented a well written discussion.